# Processing of Niobium-Alloyed High-Carbon Tool Steel via Additive Manufacturing and Modern Powder Metallurgy

**DOI:** 10.3390/ma16134760

**Published:** 2023-06-30

**Authors:** Klára Borkovcová, Pavel Novák, Nawel Merghem, Alisa Tsepeleva, Pavel Salvetr, Michal Brázda, Dragan Rajnovic

**Affiliations:** 1Department of Metals and Corrosion Engineering, University of Chemistry and Technology, Prague, Technická 5, 166 28 Prague, Czech Republic; 2Comtes FHT a.s., Průmyslová 995, 334 41 Dobrany, Czech Republic; 3Department for Production Engineering, Faculty of Technical Science, University of Novi Sad, Serbia, Trg Dositeja Obradovica 6, 21000 Novi Sad, Serbia

**Keywords:** tool steel, carbides, heat treatment, niobium

## Abstract

Niobium is recently considered one of the potential alloying elements for tool steels due to the formation of hard and stable carbides of MC type. Its use is limited by the fact that these carbides tend to coarsen during conventional melting metallurgy processing. This work explores the potential of additive manufacturing for processing Nb-alloyed tool steel with a high content of carbon. Directed energy deposition was used as the processing method. It was found that this method allowed us to obtain a microstructure very similar to that obtained after the use of consolidation via spark plasma sintering when subsequent heat treatment by soft annealing, austenitizing, oil quenching and triple tempering for secondary hardness was applied. Moreover, the soft annealing process could be skipped without affecting the structure and properties when machining would not be required. The hardness of the steel was even higher after additive manufacturing was used (approx. 800–830 HV 30) than after spark plasma sintering (approx. 720–750 HV 30). The wear resistance of the materials processed by both routes was almost comparable, reaching 5–7 × 10^−6^ mm^3^N^−1^m^−1^ depending on the heat treatment.

## 1. Introduction

Tool steels are high-tech iron-based materials that combine high hardness, high wear resistance and acceptable toughness, as well as thermal stability in the case of hot-work tools steels and high-speed steels. In the latter type, the combination of high hardness and thermal stability is achieved via specific heat treatment, which comprises austenitizing at a temperature close to 1200 °C, depending on the steel grade, quenching by oil or inert gas, and multiple (usually triple or quadruple) tempering at approx. 550 °C. The mechanism of the process is based mostly on different thermal stability of carbides and the behavior of the retained austenite [1,2,3]. Carbides in these steel grades can be divided into two main groups based on their thermal stability. Carbides with lower thermodynamic stability are M_3_C (M = Mn, Fe, Cr), M_7_C_3_ (M = Cr, Mn), M_23_C_6_ (M = Cr) and M_2_C (M = W, Mo). These carbides are characterized by lower melting points, reaching a maximum temperature of 1500 °C, and easier dissolution in austenite during austenitizing. Carbides with high thermodynamic stability have high melting points (even above 2000 °C), usually crystallize as a primary phase and are almost stable under the conditions of austenitizing. These carbides are MC (M = Ti, V, Nb), M_6_C (M = W, Mo, Fe) and M_4_C_3_ (M = V) [4,5,6].

During the secondary hardening procedure, the applied austenitizing temperature (see above) is high enough to dissolve a significant portion of M_2_C and M_7_C_3_ carbides and a low portion of MC carbides. They saturate austenite via carbon and carbide-forming elements, especially chromium, molybdenum and a lower amount of vanadium. Subsequent quenching, usually by oil or pressurized inert gas, does not allow the formation of any carbides and, hence, martensite supersaturated by these elements is obtained. Moreover, a high amount of retained austenite is formed due to the high carbon content in austenite prior to quenching. Tempering of such alloys is carried out at a high temperature, usually around 550 °C. At this temperature, precipitation of fine carbides, mostly M_6_C, MC or M_2_C types, occurs [7,8]. In higher-chromium steels, chromium-based M_7_C_3_ or M_23_C_6_ carbides could precipitate [9]. Due to related changes in austenite composition, i.e., a decrease in carbon content in austenite, which causes an increase in the “martensite start” and “martensite finish” temperatures, the retained austenite transforms to martensite upon cooling from the tempering temperature. Therefore, the tempering step is carried out three to four times in order to ensure the tempering of all additionally formed martensite in order to minimize the residual stresses and corresponding brittleness [10,11,12,13].

Even though these grades of tool steels are well-established materials, there is a continuous development of these materials. A big step forward is reached in the case of powder-metallurgy prepared using VANADIS grades of highly wear-resistant vanadium-alloyed tool steels [14]. There have also been trials to introduce other carbide-forming metals to steel, especially niobium and titanium, which are industrially applied in stainless steel to bind excessive carbon. However, their applicability in tool steels is limited because the MC carbides of these elements tend to coarsen during conventional processing route [15,16,17,18]. The process using gas atomization and hot isostatic pressing has been found to be applicable even for higher contents of these elements. Recently, we developed Nb-alloyed cold-work tool steel, with a carbon content of 2.5 wt. %. Such content normally belongs to cast irons, but in this case, the majority of carbon is bound in carbides, and hence, the material behaves like a common tool steel during processing [19]. In previous works [19,20], rapidly solidified powder of this steel, which was obtained via nitrogen gas atomization, was characterized and consolidated using hot isostatic pressing, and a heat treatment procedure was developed. It has been found that the optimal austenitizing temperature is 1100 °C because a lower temperature does not allow sufficient dissolution of carbides and, hence, the secondary hardening effect is lower, while higher temperatures cause excessive grain growth and coarsening of carbides. The precipitation of carbides during tempering at 550 °C was studied using SAED, and MC, M_2_C and M_6_C were found to be the major precipitating phases [19].

Nowadays, additive manufacturing starts to play an important role in modern industry and is more and more often considered in tooling. An example of such a technology is directed energy deposition, which works based on the principle of sprinkling powder onto a material, where the powder is sintered by means of a laser beam. In addition to powder, wire can also be used as an input material. This technology is used for the production of new components and tools or for their repair. However, the success of applying additive manufacturing via this method is rather limited to lower-carbon steels, such as AISI H13 hot-work tool steel [21]. In the case of high-carbon tool steels, there is a risk of serious problems, such as cracking. Therefore, this study tests the applicability of this additive manufacturing technology—directed energy deposition—on the structure and properties of the abovementioned high-carbon niobium-alloyed tool steel [16,22]. As a reference, spark plasma sintering was also applied.

## 2. Materials and Methods

In this work, the microstructure and mechanical properties of niobium-alloyed tool steel were determined. The chemical composition of niobium-alloyed tool steel was measured using an optical emission spectrometer, Magellan Q8 (Bruker, Billerica, MA, USA), and the results are presented in Table 1. Two compaction technologies were used. The first technology was spark plasma sintering (SPS) using an HP D10 (FCT Systeme, Rauenstein, Germany) device. Sintering was carried out at 1000 °C for 10 min, using a pressure of 80 MPa and a heating rate of 300 K/min, while the sample cooled freely in the SPS device. The diameter of the sintered sample was 20 mm. The second technology was directed energy deposition (DED) technology. The deposition of niobium-alloyed tool steel was carried out using an InssTek MX-600 machine with the SDM800 module (the spot size of the laser was 800 μm, the layer height was 250 μm, and the hatch distance was 500 μm). The printing speed was 850 mm/min with a powder feed rate of 3 g/min. All samples were printed in the DMT mode (a mode for automatic laser power adjustment to maintain the layer height) with laser power ranging from 400 to 600 W.

After compaction, the samples were further heat-treated. During compaction using spark plasma sintering (SPS) or direct energy deposition (DED), soft annealing in an electric resistance furnace (Martinek, Kladno, Czech Republic) at 780 °C for 8 h was used; then, the samples were removed from the furnace to cool down and was placed inside for another 2 h at 680 °C. After that, austenitization at 1100 °C for 30 min and quenching in oil were carried out. The quenched samples were triple tempered at 550 °C for 1 h, followed by air cooling. As stated above, the heat treatment was optimized using HIP-consolidated samples recently [19]. One sample in the printed state was subjected to heat treatment without soft annealing.

Because our experience has shown that X-ray diffraction analysis does not reveal carbides other than M_7_C_3_ in this alloy [19], selective etching was used to highlight individual carbides according to the procedure described in [23]. Etching to highlight MC-type carbides, i.e., mainly NbC carbides, was carried out electrolytically in 10 g of CrO_3_ in 50 mL of water. Chemical etching in 2 g of KMnO_4_, 2 g of NaOH, and 50 mL of water reveals carbides of the M_2_C and M_6_C types. Using the same solution and electrolytic etching, carbides M_2_C, M_6_C and M_7_C_3_ were revealed. From the selective carbide etching, the area fraction, carbide size, and Feret’s diameters were determined using the thresholding operations of the ImageJ software.

After etching the samples in a Nital solution (10 mL of nitric acid and 90 mL of ethanol), their microstructure was observed using a scanning electron microscope (SEM) VEGA 3 LMU (TESCAN, Brno, Czech Republic). The scanning electron microscope was equipped with an energy-dispersive spectrometer (EDS) with an X-max 20 mm^2^ detector (Oxford Instruments, High Wycombe, UK).

In addition to the phase analysis and microstructure evaluation, the samples were characterized from the viewpoints of mechanical properties (hardness), as well as tribological behavior (wear rate and friction coefficient). Hardness was measured using the Vickers method at a load of 30 kg (HV 30), which corresponds to 294 N. Hardness was measured ten times for each sample and the average value was calculated. Wear resistance was measured using a TriboTester tribometer (Tribotechnic, Clichy, France) with the ball-on-disc tribometer in a linear reciprocal mode (*e* excenter, i.e., the length of one sliding motion, of 5 mm). In this case, the “ball” was made of alumina (α-Al_2_O_3_) and it was 6 mm in diameter, and the “disc” was a sample. A schematic of the wear test is shown in Figure 1 below. No lubricant was used during the process. The sliding distance (*l*) in this process was 20 m and a normal force (*F*) of 5 N was used. The wear rate (*w* [mm^3^ N^−1^ m^−1^]) was calculated using Equation (1), wherein we take into account the wear track section area (*A* [mm^2^]).
(1)$$w=\frac{A\cdot e}{F\cdot l}$$

In addition to the wear rate, the friction coefficient was recorded during the test via the TriboTester proprietary software (Tribotechnic, Clichy, France). The calculation of the friction coefficient (*f*, dimensionless) is based on Equation (2):(2)$$f=\frac{F_{f}}{F}$$
where *F_f_* means the friction force [N], which was measured using the force sensor of the TriboTester tribometer, and *F* is the normal force defined above.

## 3. Results

### 3.1. Microstructure

The microstructure of the spark plasma sintered material in the as-sintered state is composed of various types of carbides embedded in an iron-based matrix (see Figure 2). The almost rounded particles, which appear darker than the matrix in the SEM-BSE image, are chromium-rich M_7_C_3_ carbides [24]. The coarse light ones contain niobium and molybdenum (Figure 3), while the fine ones are composed of niobium, molybdenum and vanadium. These carbides are, hence, of MC type [24]. Their chemical composition is presented below in Table 2. The above assignment of the phases is based on experience with determining their microstructure using a combination of chemical analysis and SAED in a previous study using a material processed by HIP, which was published recently [24]. 

Subsequent heat treatment, which included soft annealing, austenitizing, oil quenching and tempering, resulted in a very similar structure (Figure 4), where slight coarsening of carbides occurred. The distribution of elements (Figure 5) in the sample did not undergo any significant change, except for the presence of carbides with a high amount of tungsten (probably of M_6_C type). These carbides are very likely present in all states; but due to their low area fraction (see below), they are not easy to be detected. Moreover, these carbides can contain either tungsten or molybdenum, which can almost freely substitute mutually. The phase composition is presented in Table 3.

The material processed using DED in the as-deposited state is characterized by the presence of a carbidic network, which is typical for ledeburitic tool steels in the as-cast state (Figure 6). This network, composed of eutectic carbides, is morphologically similar to ledeburite in cast irons, and therefore, these steels are called “ledeburitic“. This network is composed mostly of chromium carbides (Figure 7). The lighter particles dispersed in the matrix are MC carbides rich in vanadium, niobium and molybdenum (Table 4).

When such material was processed by heat treatment, including soft annealing, austenitizing, oil quenching and tempering according to the conditions defined in a recent study [24] and listed above, the carbide network disintegrated into separate carbide particles (Figure 8). The resulting structure is very similar to the structure previously published [24] or the SPS-processed one, including the size of carbides. This confirms the applicability of DED in the processing of this grade of tool steel. The phase composition of this sample is presented in Table 5. The distribution of the elements is shown in Figure 9.

The material in the printed state after heat treatment without soft annealing was determined in order to see if the carbide network would disintegrate in the same way as during the complete heat treatment procedure. As shown in Figure 10, the distribution and the size of carbides are comparable to the above-described state. The phase composition of the sample (Table 6) and the distribution of elements (Figure 11) are also very similar to the sample presented above. 

The results (see Table 7) show that the average size of the observed carbide particles is not significantly affected by the heat treatment, i.e., coarsening did not occur during the selected heat treatment procedure. Also, there is no systematic dependence of the area fraction of MC, M_2_C and M_6_C carbides after the heat treatment procedure. M_6_C and M_2_C are present in a lower amount, being probably the areas with higher tungsten and/or molybdenum contents in the EDS maps above.

### 3.2. Mechanical and Tribological Properties

The mechanical and tribological properties are shown in Table 8. The dependence of the friction coefficient on time is captured in Figure 12. The wear resistance test showed that the sintered samples exhibit the lowest value of friction coefficient, which means they have the best sliding properties, whereas the samples in as-deposited state show the highest values. On the other hand, the friction coefficient values are more stable with time for the printed samples than the sintered ones, especially in the state after the heat treatment without soft annealing. These variations are probably caused by the higher hardness of the printed samples in the heat-treated state (Table 8). The sintered sample after heat treatment is more resistant to wear than the sample in as-sintered state, as indicated by the wear resistance values in Table 8. The DED-processed samples reach the highest wear resistance in the as-printed state, probably due to rapid cooling during the printing process, which formed a very fine structure and non-tempered martensite in the steel. However, such a state will probably be too brittle for practical use. It is also clear that skipping the soft annealing process when manufacturing the DED-processed tool steel leads to an improvement in wear resistance (see Table 8), probably due to the presence of slightly finer carbides.

## 4. Discussion

Niobium forms stable MC carbides, as proven recently [24,25] and in this work. These carbides are highly stable and, therefore, they do not dissolve significantly during austenitizing. Thus, it could be concluded that they do not strongly affect the secondary hardening effect. In reality, the presence of niobium increases the hardness of steel in all states because of the stability of these carbides throughout the manufacturing and processing of steel. It is advantageous for the tempered state, but not for the soft-annealed one, because it lowers machinability. Hence, processing via additive manufacturing or other net-shape or near net-shape processing of Nb-alloyed steels could be very advantageous. 

There are two contradictory phenomena, which influence the applicability of additive manufacturing processes in the case of Nb-alloyed high-carbon tool steels:A high content of niobium implies the need for high solidification rates in order to suppress the coarsening of MC carbides.A high carbon content, together with a high amount of chromium in steel, increases the quenching ability of steel. Therefore, martensite can form during the cooling of 3D printed products, which could lead to cracking of the products.

It implies that the cooling rate has to be balanced between these limits. When we compare selective laser melting (SLM) and directed energy deposition (DED), we can conclude that DED usually reaches a lower cooling rate. This work confirmed the suitability of DED because cracking was not observed, even though very high hardness and wear resistance were achieved even in the as-printed state. The fact that the network of carbides was disintegrated during heat treatment is also very interesting and important. In order to prove whether austenitizing is sufficient for the disintegration of the carbide network, heat treatment without soft annealing was tested (see above) and proved to be successful. It implies that the selected austenitizing temperature (1100 °C), which was optimized based on previous work [24], is high enough to dissolve a significant portion of chromium-based M_7_C_3_ carbides. Therefore, in the case of 3D printing processes when machining is not required, soft annealing can be skipped, which lowers the energy consumption and carbon footprint of the material used. This fact can highly likely be generalized to all steels of a similar type, i.e., high-carbon “ledeburitic” tool steels.

## 5. Conclusions

This work proved the applicability of DED for processing high-carbon Nb-alloyed tool steel. Subsequent heat treatment using two-step soft annealing, austenitizing at 1100 °C, oil quenching and triple tempering at 550 °C resulted in the same type of microstructure as obtained using spark plasma sintering and the same heat treatment. The hardness and wear resistance of the additively manufactured material was slightly higher than the SPS-processed one, reaching approx. 800–830 HV 30 and 5–7 × 10^−6^ mm^3^N^−1^m^−1^, respectively. In the case of additive manufacturing of this steel, soft annealing can be skipped, thus saving energy and lowering the carbon footprint.

## Figures and Tables

**Figure 1 materials-16-04760-f001:**
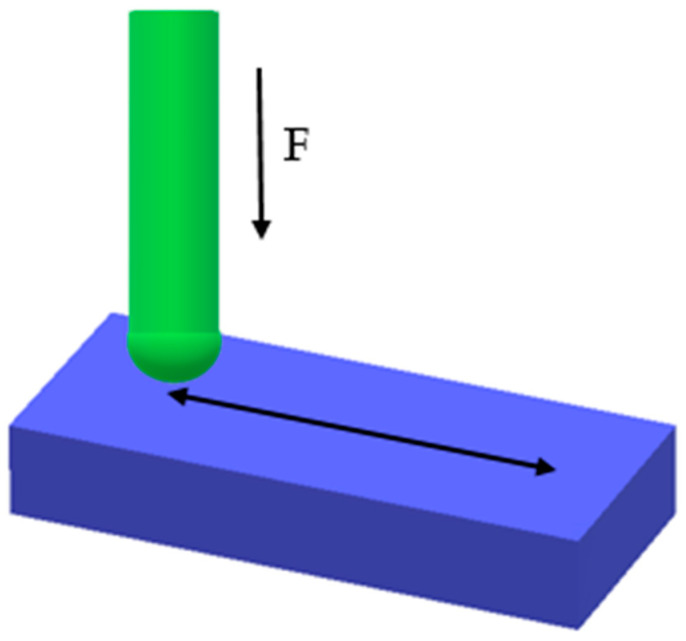
A schematic of the wear test.

**Figure 2 materials-16-04760-f002:**
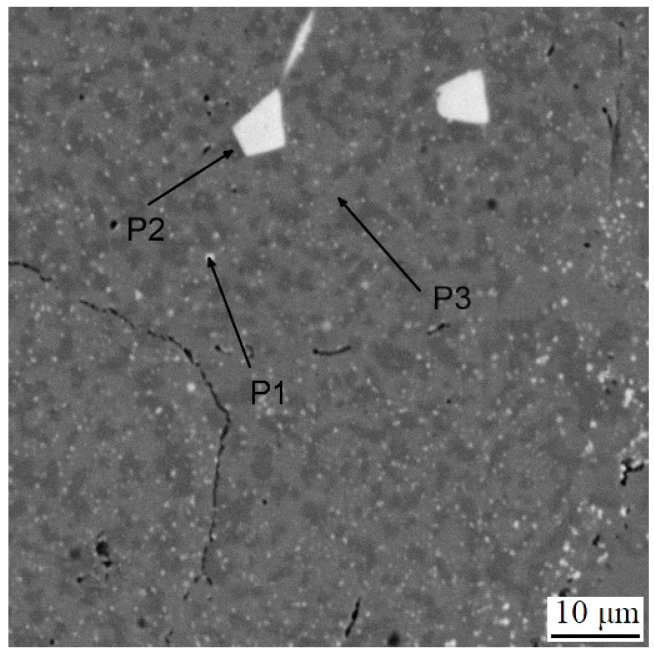
Microstructure of the sample in the as-sintered state. P1–P3 mean the points analyzed by EDS, as presented in Table 2.

**Figure 3 materials-16-04760-f003:**
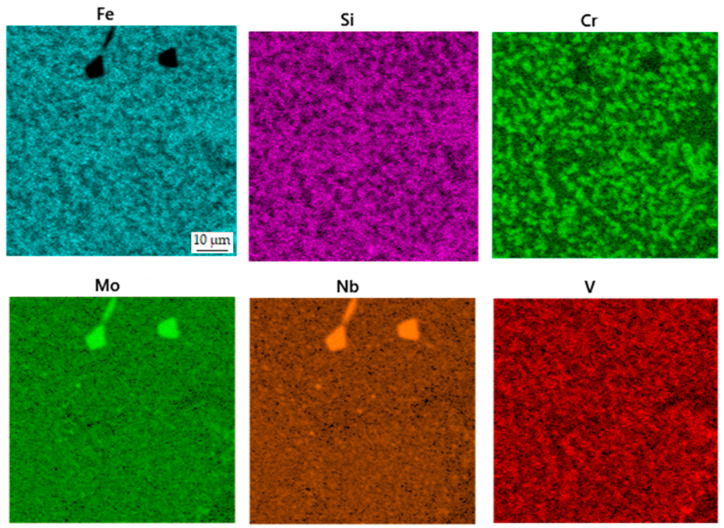
EDS map of the distribution of elements in the sample in the as-sintered state.

**Figure 4 materials-16-04760-f004:**
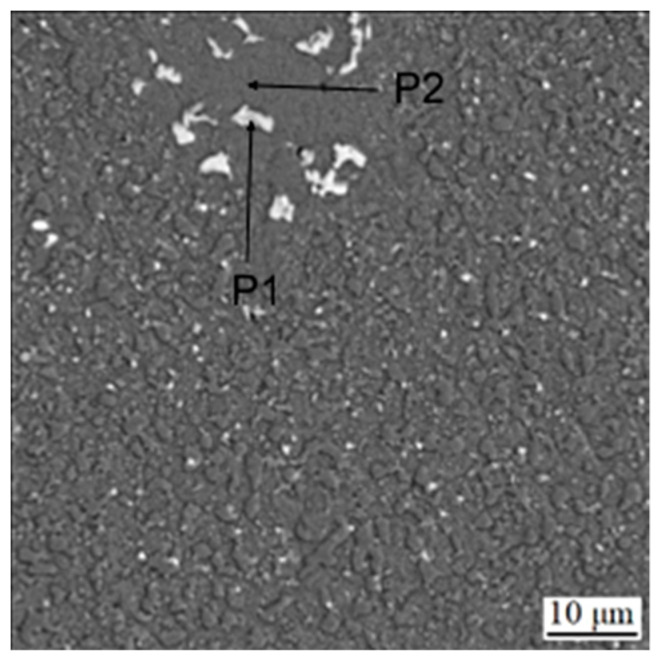
Microstructure of the sample in the sintered state after heat treatment. P1 and P2 mean the points analyzed by EDS, as presented in Table 3.

**Figure 5 materials-16-04760-f005:**
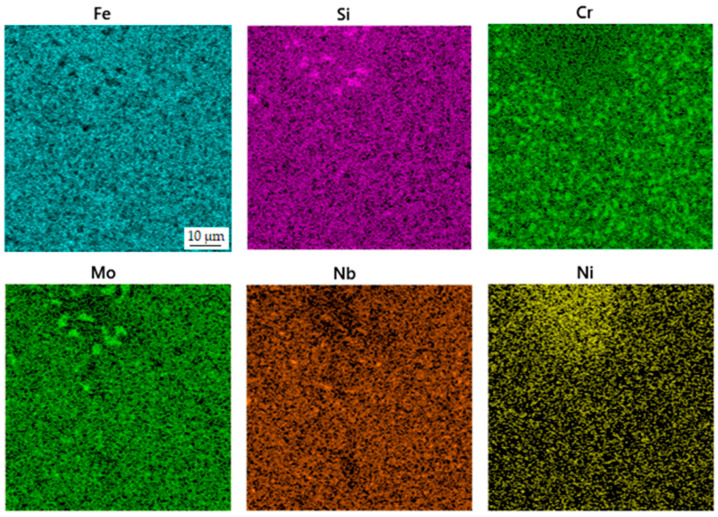
EDS map of the distribution of elements in the sintered sample after heat treatment.

**Figure 6 materials-16-04760-f006:**
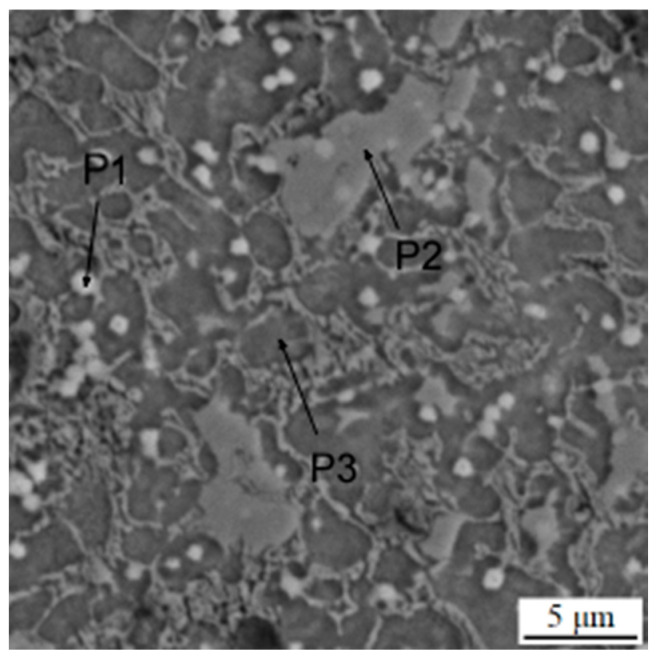
Microstructure of the sample in the as-printed state. P1–P3 mean the points analyzed by EDS, as presented in Table 4.

**Figure 7 materials-16-04760-f007:**
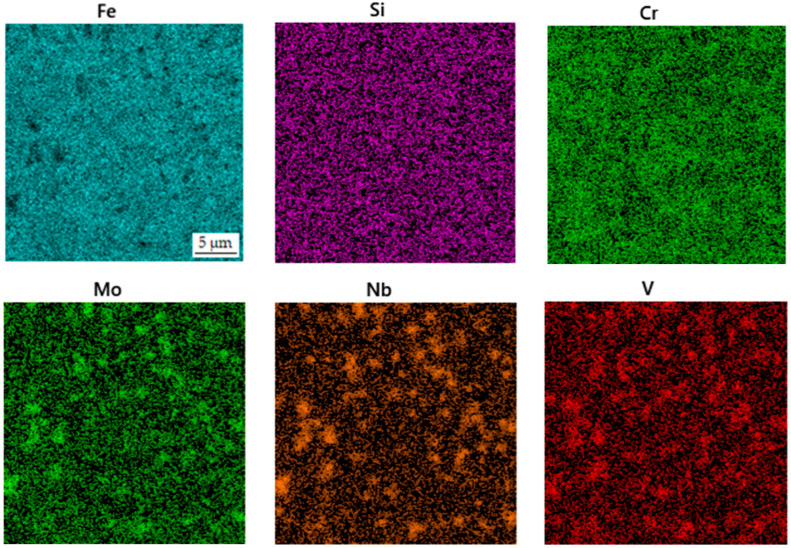
EDS map of the distribution of elements in the sample in the as-printed state.

**Figure 8 materials-16-04760-f008:**
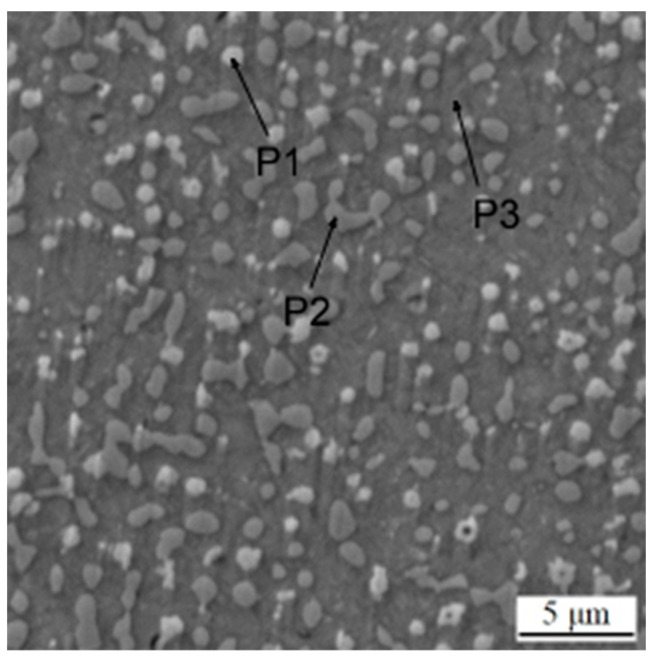
Microstructure of the printed sample after heat treatment. P1–P3 mean the points analyzed by EDS, as presented in Table 5.

**Figure 9 materials-16-04760-f009:**
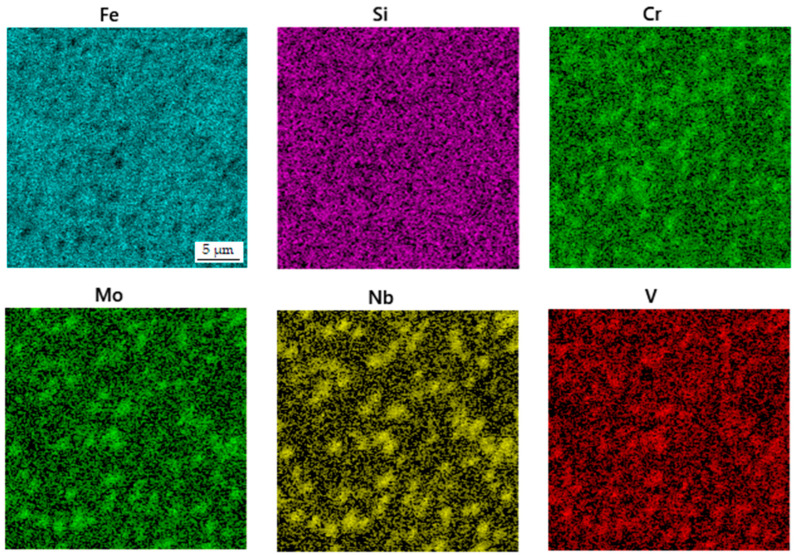
EDS map of the distribution of elements in the printed sample after heat treatment.

**Figure 10 materials-16-04760-f010:**
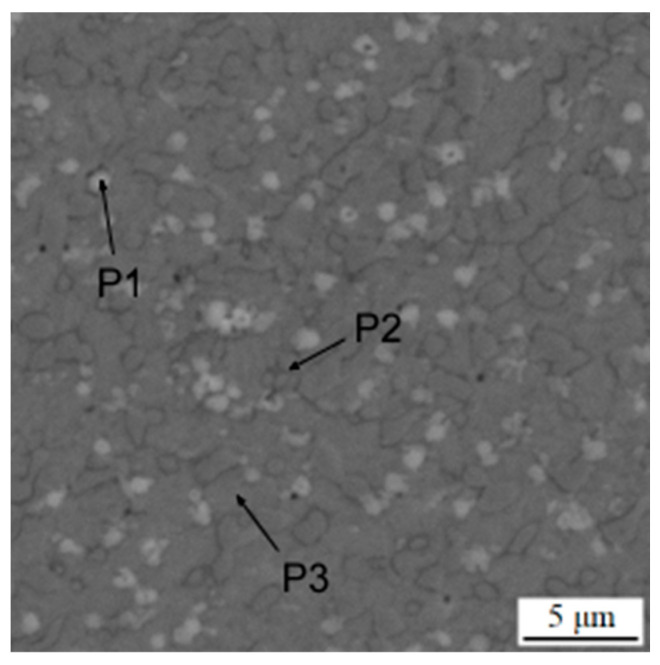
Microstructure of the sample in the printed state after heat treatment without soft annealing. P1–P3 mean the points analyzed by EDS, as presented in Table 6.

**Figure 11 materials-16-04760-f011:**
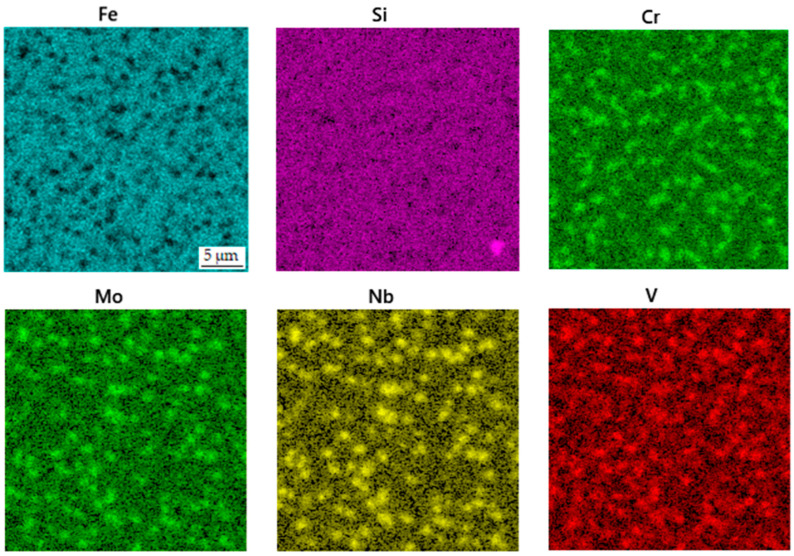
EDS map of the distribution of elements in the sample in the printed state after heat treatment without soft annealing.

**Figure 12 materials-16-04760-f012:**
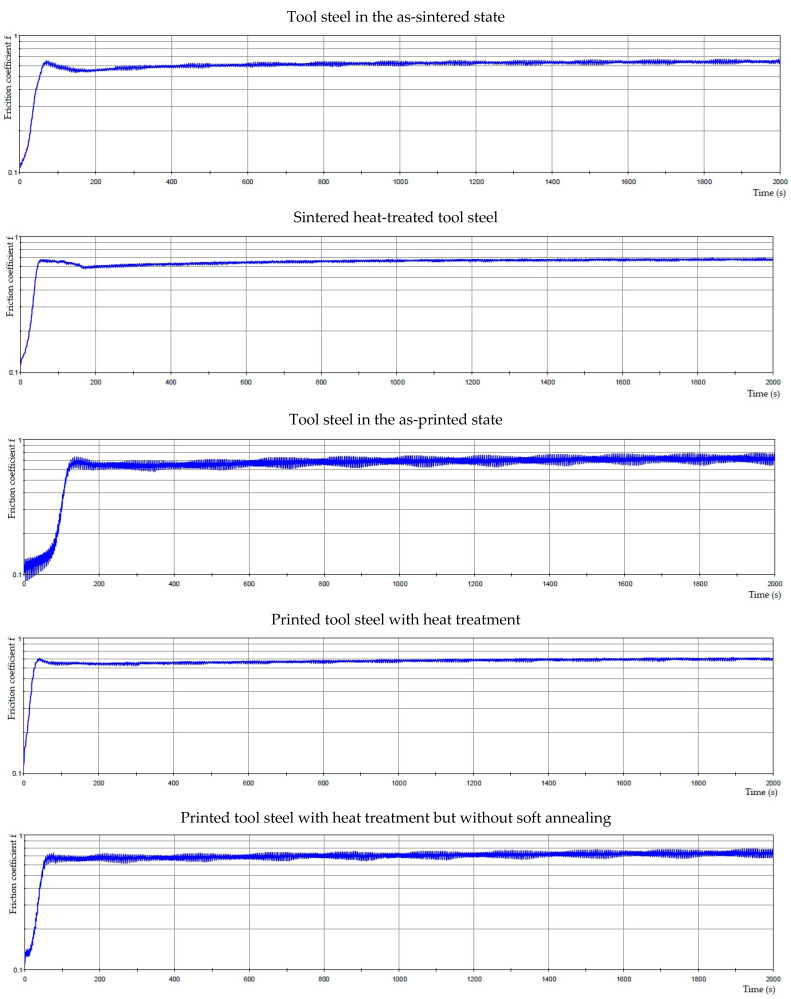
The friction coefficient of tool steels depends on time.

**Table 1 materials-16-04760-t001:** Chemical composition of niobium-alloyed tool steel (in wt. %).

	Fe	C	Cr	Nb	V	W	Si	Ni	Mo
Tool steel	bal.	2.5	6.2	3.0	2.6	1.0	3.3	1.4	2.2

**Table 2 materials-16-04760-t002:** Phase composition of the sample in the as-sintered state.

	Fe	Nb	C	V	W	Cr	Mo	Si	Ni
P1 wt. %	16.6	46.0	16.4	9.6	2.1	2.2	3.9	0.6	0.5
P2 wt. %	59.2	1.7	11.9	5.3	-	16.0	3.3	-	-
P3 wt. %	79.8	1.4	5.5	1.5	-	2.7	0.9	4.3	3.2

**Table 3 materials-16-04760-t003:** Phase composition of the sintered sample after heat treatment.

	Fe	Mo	W	C	V	Cr	Co	Si	Ni
P1 wt. %	35.0	22.9	11.7	8.1	2.9	5.6	-	5.2	8.7
P2 wt. %	76.5	0.7	-	5.6	0.4	3.1	1.0	4.0	8.3

**Table 4 materials-16-04760-t004:** Phase composition of the sample in the as-printed state.

	Fe	Nb	W	C	V	Cr	Mo	Si	O
P1 wt. %	23.1	36.3	1.6	18.0	11.4	5.2	3.6	0.9	-
P2 wt. %	80.3	1.8	-	5.8	2.0	5.3	1.0	3.4	-
P3 wt. %	75.7	8.7	-	8.7	1.5	6.1	-	-	3.5

**Table 5 materials-16-04760-t005:** Phase composition of the printed sample after heat treatment.

	Fe	Nb	C	V	Cr	Mo	Si	W
P1 wt. %	45.1	22.7	9.8	9.6	4.2	4.3	1.9	2.0
P2 wt. %	76.6	1.2	9.6	1.9	4.9	1.9	3.5	-
P3 wt. %	55.7	12.0	9.0	9.4	5.0	4.2	2.9	1.9

**Table 6 materials-16-04760-t006:** Phase composition of the sample in the printed state after heat treatment without soft annealing.

	Fe	Nb	C	V	Cr	Mo	Si	W
P1 wt. %	58.2	12.0	10.4	6.7	4.6	3.3	2.8	1.5
P2 wt. %	74.3	4.5	6.3	3.1	4.6	2.0	3.5	0.9
P3 wt. %	80.6	0.7	6.3	1.0	5.2	1.4	3.8	-

**Table 7 materials-16-04760-t007:** Area fraction of the samples and their carbides’ size.

	MC Carbide		M_6_C and M_2_C Carbides	
	Area Fraction (%)	Average Diameter (µm)	Area Fraction (%)	Average Diameter (µm)
Tool steel in the as-sintered state	19	1.8	8	0.8
Tool steel in the as-printed state	30	2.5	9	1.3
Printed tool steel with heat treatment	25	2.3	10	1.2
Printed tool steel with heat treatment but without soft annealing	29	2.5	9	1.4

**Table 8 materials-16-04760-t008:** Mechanical and tribological properties of the tested tool steels against Al_2_O_3_ ball. f—friction coefficient; w—wear rate.

Tool Steel	Hardness (HV 30)	Ra (µm)	f (Al_2_O_3_)	w (Al_2_O_3_) (mm^3^N^−1^m^−1^)
Tool steel in the as-sintered state	751 ± 41	0.0133	0.58	$6.28\times 10^{-6}$
Sintered heat-treated tool steel	719 ± 5	0.0156	0.55	$4.78\times 10^{-6}$
Tool steel in the as-printed state	729 ± 19	0.0193	0.67	$5.20\times 10^{-6}$
Printed tool steel with heat treatment	828 ± 8	0.0322	0.65	$7.13\times 10^{-6}$
Printed tool steel with heat treatment but without soft annealing	803 ± 22	0.0315	0.68	$6.20\times 10^{-6}$

## Data Availability

Not applicable.

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
