# Peer review of "Processing of Niobium-Alloyed High-Carbon Tool Steel via Additive Manufacturing and Modern Powder Metallurgy"

_materials, 2023, doi:10.3390/ma16134760_

Round 1

Reviewer 1 Report

The submitted manuscript explores the potential of additive manufacturing for processing of Nb-alloyed tool steel with high content of carbon. Directed energy deposition was used as the processing method. It was found that this method allows to reach very similar microstructure as the use of consolidation by spark plasma sintering, when subsequent heat treatment by soft annealing, austenitizing, oil quenching and triple tempering for secondary hardness was applied. Moreover, the soft annealing could be skipped without affecting the structure and properties, when machining will not be required. The resulting hardness and wear resistance do not differ significantly between additive manufacturing and spark plasma sintering consolidation. The subject matter is important and has a special value considering the practical applications, especially, during rocessing of niobium-alloyed high-carbon tool steel by additive manufacturing and modern powder metallurgy. The paper is rather clearly presented and well organized. The references are adequate. The conclusions seem to be sound and justified. However, I suggest a mandatory revision regarding the following points to increase the quality of the paper:

1)     What does the "e" in equation (1) mean? It is important for the wear rate calculation.

2)     In the section 2. Materials and Methods the scheme of the wear test, including the positions of the sample and counter-sample, should be provided.

3)     The diagrams of friction wear coefficients vs.. the time of friction should be provided for all the samples produced.

4)     The method of calculation of friction wear coefficients should be provided.

5)     English should be carefully checked and corrected,.

Minor editing of English language required.

Author Response

  • What does the "e" in equation (1) mean? It is important for the wear rate calculation.

The meaning of "e" is excenter. It is the distance of one sliding motion. It is now explained in the text.

  • In the section 2. Materials and Methods the scheme of the wear test, including the positions of the sample and counter-sample, should be provided.

The scheme was added as new Figure 1.

  • The diagrams of friction wear coefficients vs.. the time of friction should be provided for all the samples produced.

The diagrams were added as new Figure 12.

  • The method of calculation of friction wear coefficients should be provided.

The equation (2) and corresponding explanation were added.

  • English should be carefully checked and corrected.

English was corrected and minor revisions were carried out.

Reviewer 2 Report

The article is interesting, but seems to be very general, and it should be further developed.

Abstract: Provide some numerical values as properties. It seems general.

Introduction: This part of the paper is poorly written, at a high level of generality. The achievements of other authors in this field should be presented with more precision. The authors write general sentences, for example: […] " In the latter type, the combination of high hardness and thermal stability is achieved by specific heat treatment, which is based mostly on different thermal stability of carbides [1-3]."; […] Therefore, the tempering step is carried out three to four times in order to ensure tempering of all additionally formed martensite [7-10]”. There are more such examples in this section of the article. It would be good to provide more details.

You can try to include more works cited in the article. The introduction is developed only on the basis of 17.

The novelty of the work needs to be highlighted more.

Materials and Methods: Why was a temperature of 1100°C used for austenitising, when beyond this temperature the effect of dissolving M6C carbides can be achieved, which can improve the effect of secondary hardness on tempering? Please comment.

If there is a shadow of doubt that M6C carbides are present in the material, a temperature close to the solidus line should be used as a criterion for selecting the temperature for austenitizing, to obtain the greatest possible enrichment of the matrix in alloying elements other than chromium, which in this type of steel is rather present in the form of Cr23C6 carbides. For this purpose, X-ray studies are necessary to identify them.

Results:

What is the density of the tested materials. The results of the density measurement should be included in the article.

 X-ray phase analysis results are definitely missing from the paper. The article needs to be supplemented in this regard. The analysis based on the SEM/EDS results is insufficient.

The article is very general and lacks in-depth analysis. A more in-depth analysis is recommended.

It is recommended to summarise the hardness results in the form of a table.

Conclusions: Conclusions are at a high level of generality; it is recommended that the numerical data obtained be taken into account in the conclusions.

Author Response

Abstract: Provide some numerical values as properties. It seems general.

The numerical data were added to Abstract.

Introduction: This part of the paper is poorly written, at a high level of generality. The achievements of other authors in this field should be presented with more precision. The authors write general sentences, for example: […] " In the latter type, the combination of high hardness and thermal stability is achieved by specific heat treatment, which is based mostly on different thermal stability of carbides [1-3]."; […] Therefore, the tempering step is carried out three to four times in order to ensure tempering of all additionally formed martensite [7-10]”. There are more such examples in this section of the article. It would be good to provide more details.

The corresponding text was improved.

You can try to include more works cited in the article. The introduction is developed only on the basis of 17.

More papers were cited.

The novelty of the work needs to be highlighted more.

The prime novelty description was improved (“However, the success in additive manufacturing by this method is rather limited to lower-carbon steels, such as AISI H13 hot-work tool steel [21]. In the case of high-carbon tool steels, there is a risk of serious problems, such as cracking. Therefore, this paper tests the applicability of this additive manufacturing technology – directed energy deposition – on the structure and properties of the above mentioned high-carbon niobium-alloyed tool steel [16,22]. As a reference, spark plasma sintering was also applied.”).

Materials and Methods: Why was a temperature of 1100°C used for austenitising, when beyond this temperature the effect of dissolving M6C carbides can be achieved, which can improve the effect of secondary hardness on tempering? Please comment.

The conditions of heat treatment were defined in our recent work for the same composition of the steel, but processed by different method (consolidation by HIP). Higher temperatures than 1100 °C caused excessive grain growth, as well as the coarsening of carbides. The explanation was added to the text.

If there is a shadow of doubt that M6C carbides are present in the material, a temperature close to the solidus line should be used as a criterion for selecting the temperature for austenitizing, to obtain the greatest possible enrichment of the matrix in alloying elements other than chromium, which in this type of steel is rather present in the form of Cr23C6 carbides. For this purpose, X-ray studies are necessary to identify them.

The XRD analysis was used in our recent work (ref. [24]), which recognized martensite and M7C3 carbide only. For the identification of other types, SAED in TEM and selective etching were found to be efficient. Therefore we used selective etching in this work.

Results:

What is the density of the tested materials. The results of the density measurement should be included in the article.

The porosity of the samples was studied by the image analysis in the non-etched state and it was found to be below the detection limit (< 0.5 area %) for both processing routes.

 X-ray phase analysis results are definitely missing from the paper. The article needs to be supplemented in this regard. The analysis based on the SEM/EDS results is insufficient.

The XRD analysis was used in our recent work (ref. [24]), which recognized martensite and M7C3 carbide only. The reason is in overlaps of the carbides’ diffraction lines and their low amounts. Therefore, we didn’t use it in this work.

The article is very general and lacks in-depth analysis. A more in-depth analysis is recommended.

The analysis of the results was improved.

It is recommended to summarise the hardness results in the form of a table.

Hardness is now in Table 8 and discussed in more details.

Conclusions: Conclusions are at a high level of generality; it is recommended that the numerical data obtained be taken into account in the conclusions.

Numerical data were added to Conclusions.

Reviewer 3 Report

The article can be accepted for publication with some additional data that allow us to trace the entire evolution of the microstructure and phase composition of a niobium-alloyed high-carbon tool steel: from the powder to additive manufacturing steel. Below are some comments on the manuscript.

1. The paper provides information on the chemical composition of niobium-alloyed steel, but there is no information about the powder used to produce steel using spark plasma sintering and additive manufacturing, i.e. the method of obtaining the powder, shape and size, its microstructure and phase composition. The thermal stability and distribution of niobium carbides are of the interest during the heat treatment of steel as well as during the melting and rapid crystallization of the powder. It is also interesting how the hardness of the steel changes in powder form and after its consolidation.

2. Why don't you use the most reliable and simple method for analyzing the phase composition of steel, i.e. XRD analysis? The given EDS maps can only show the distribution of elements in individual zones of the samples and indirectly identify the phase composition. Can we trust these data? Does the entire specimen have the same phase composition throughout its volume? There may be phases in the alloy that are smaller than the area scanned in the EDS analysis.

Author Response

  1. The paper provides information on the chemical composition of niobium-alloyed steel, but there is no information about the powder used to produce steel using spark plasma sintering and additive manufacturing, i.e. the method of obtaining the powder, shape and size, its microstructure and phase composition. The thermal stability and distribution of niobium carbides are of the interest during the heat treatment of steel as well as during the melting and rapid crystallization of the powder. It is also interesting how the hardness of the steel changes in powder form and after its consolidation.

The powder was studied in recent work of our team. The reference and explanation were added.

  1. Why don't you use the most reliable and simple method for analyzing the phase composition of steel, i.e. XRD analysis? The given EDS maps can only show the distribution of elements in individual zones of the samples and indirectly identify the phase composition. Can we trust these data? Does the entire specimen have the same phase composition throughout its volume? There may be phases in the alloy that are smaller than the area scanned in the EDS analysis.

The XRD analysis was used in our recent work (ref. [24]), which recognized martensite and M7C3 carbide only. The reason is in overlaps of the carbides’ diffraction lines and their low amounts. Therefore, we didn’t use it in this work.

Round 2

Reviewer 1 Report

The authors have worked on the comments suggested by the Reviewers. In my opinion th manuscript can be published.

Reviewer 2 Report

Thank you for responding to my comments and suggestions. I accept the article in the present form.

Reviewer 3 Report

Authors replied to all my remarks, and I recommend this article to be published